# The Effects of Tea Polyphenol on Chicken Protein Digestion and the Mechanism under Thermal Processing

**DOI:** 10.3390/foods12152905

**Published:** 2023-07-31

**Authors:** Wenjun Wen, Shijie Li, Junping Wang

**Affiliations:** 1College of Food Science and Engineering, Shanxi Agriculture University, Shanxi 030801, China; wenwenjun@sxau.edu.cn; 2Medical College, Nankai University, Tianjin 300350, China; 9820200024@nankai.edu.cn; 3State Key Laboratory of Food Nutrition and Safety, Tianjin University of Science & Technology, Tianjin 300457, China

**Keywords:** chicken protein, tea polyphenol, digestion, protein structure, protease activity

## Abstract

Meat product is the main food and major source of daily protein intake. Polyphenols are always introduced into many meat products during processing. Some complex interactions may occur between polyphenol and meat protein during the processing, especially thermal processing, which may affect the digestion of protein. In this experiment, chicken protein and tea polyphenol were interacted in simulated systems to explore the effects of the interaction between meat protein and polyphenols on the digestion of meat protein. The mechanism of tea polyphenol inhibiting chicken protein digestion was studied by analyzing the changes of chicken protein in intrinsic fluorescence, surface plasmon resonance (SPR), reactive sulfhydryl group, and solubility in different solvents. The results showed that the chicken protein digestion had a negative correlation with tea polyphenol concentration and interaction temperature, and the meat protein has a higher affinity to EGCG than protease. The mechanism of tea polyphenol inhibiting chicken protein digestion was related to the changing spatial structure of chicken protein and the decreasing activity of proteases. In the simulation system, at low-concentration tea polyphenol, the inhibition of the tea polyphenol on the digestibility of chicken protein might be mainly caused by the changes in chicken protein structure, while at high concentration, the changes in protein structure and the inhibition of proteases activity played a role together. This experiment revealed the effect and the mechanism of polyphenols on the digestion performance of meat protein and provide more references for the further application of polyphenols in meat processing.

## 1. Introduction

Meat product is the main food and a major source of daily protein intake, and many meat products have been developed to meet consumption needs. Condiment is a necessary ingredient during the processing of meat products, and many condiments contain high levels of polyphenols [1]. Moreover, the polyphenol as a kind of effective antioxidant is usually used to protect the protein from oxidation-induced deterioration and increase shelf-life [2]. So, a series of complex reactions may occur between polyphenols and proteins during processing and storage, especially under high-temperature conditions during thermal processing, which not only significantly affect the texture, flavor, and shelf life of meat products, but also possibly affect the protein digestion [3,4,5]. There is plenty of literature reporting the interactions between meat protein and polyphenols, meat proteins from pork [6], fish [7], sardine [8], mackerel [9], and cuttlefish [10]; polyphenols including catechin [11], quercetin [12], and resveratrol [13]. These studies have focused on investigating the effects of polyphenols on the function of meat proteins (gelation, water-holding capacity, and texture properties) [14] and the antioxidant capacity of polyphenols during the oxidation of meat proteins [15,16]. 

As an important nutrient, the efficient absorption and utilization of protein is very important for human health. In a daily diet, the reaction between polyphenols and meat protein is inevitable. While we study the functional properties of meat protein during the interaction between polyphenols and meat protein, it is necessary to further discuss whether the interaction affects the nutritional efficacy of protein. Chicken is a common edible meat, and it occupies a high proportion in meat consumption because of its high protein content and low price. Tea polyphenol as one of the most important functional components in tea is also widely studied and applied in the food industry (lipid antioxidation, meat products preservation, functional foods, etc.) [17]. The effects of tea polyphenols on the digestibility of chicken myofibrillar protein in an enhanced oxidation system were investigated. But the interaction between chicken protein and tea polyphenol at thermal processing has not been investigated [18]. Therefore, chicken protein (CP) and tea polyphenol (TP) were chosen to explore the interaction between meat protein and polyphenol and the effects on meat protein digestion during thermal processing.

The interaction between chicken protein and tea polyphenol was simulated in three modes, which include interaction between native chicken protein and tea polyphenol, interaction between heated chicken protein and tea polyphenol, and interaction between chicken protein and tea polyphenol at different temperatures. The digestion performance of the complex of chicken protein and tea polyphenol was characterized by digestibility and the peptide map after digestion. Intrinsic fluorescence spectra and reactive sulfhydryl groups were employed to characterize the variation of protein spatial structure, and the surface plasmon resonance (SPR) was applied to analyze the affinity between tea polyphenols and protein. The purpose of this experiment is to reveal the effect of polyphenols on the digestion performance of meat protein and the mechanism to provide more references for the further application of polyphenols.

## 2. Materials and Methods

### 2.1. Materials

The frozen chicken breast was purchased from the local market. The tea polyphenol from green tea was purchased from Shanghai Yuanye Bio-Technology Co., Ltd, Shanghai, China. The water used in this study was ultrapure water. Myoglobin, myosin, and proteases, including pepsin (>2500 U/mg), trypsin (1655 U/mg), and chymotrypsin (>40 U/mg)from porcine, were purchased from Sigma-Aldrich Chemical Corporation, St. Louis, MO, USA. (−)-Epigallocatechin gallate (EGCG) was obtained from Macklin Biochemical Corporation, Shanghai, China. The ZipTip C18 pipette tip for sample purification was purchased from Merck Millipore Ltd, Darmstadt, Germany.

### 2.2. Sample Preparation

After all visible fat and connective tissue were removed, the chicken breast was ground to mince. The meat mince was defatted by n-hexane twice, then lyophilized and ground to powder, which was the native chicken protein ((n)CP). The native chicken protein powder was heated at 100 °C for 10 min to obtain the heated chicken protein((h)CP). 

The CP-TP (complex of chicken protein and tea polyphenol) was acquired according to different interaction modes and resulted in three complexes.

(a) (n)CP-TP (complex of native chicken protein and tea polyphenol): 50 mg native chicken protein added into 1 mL of tea polyphenol at concentrations of 0, 0.1, 0.5, 1, 2, 5, 10, and 20 mg/mL, mixed for 10 min. 

(b) (h)CP-TP (complex of heated chicken protein and tea polyphenol): 50 mg heated chicken protein added into 1 mL of tea polyphenol at concentrations of 0, 0.1, 0.5, 1, 2, 5, 10, and 20 mg/mL, mixed for 10 min.

(c) CP-TP(h) (complex of chicken protein and tea polyphenol heated together): 50 mg native chicken protein and 1 mL of tea polyphenol at concentrations of 0, 0.1, 0.5, 1, 2, 5, 10, and 20 mg/mL heated together at 100 °C for 10 min. 

Four different tea polyphenol concentrations (0, 0.1, 1, and 10 mg/mL) of CP-TPs were processed at different temperatures (20, 40, 60, 80, and 100 °C) to investigate the effects of temperature on CP-TP digestion. CP-TPs were heated in a water bath with magnetic stirring for 10 min, especially the mixture that was treated at 20°C in a constant temperature incubator. 

### 2.3. Simulated Digestion of Sample

The simulated digestive system was modified according to our early work [19], including 1 h simulated gastric digestion and 4 h simulated intestinal digestion. The digestion was stopped by a 5 min boiling water bath. After cooling to room temperature, the products were centrifuged and the supernatants were collected to quantify the degree of hydrolysis (DH).

### 2.4. Determination of Sample Digestibility

The method of DH determination was optimized according to Wu et al. [20]. The supernatants collected from the samples as mentioned in Section 2.3 were diluted 20 times with 100 mM sodium bicarbonate, then mixed with OPA reagent at 1:1 (*v*/*v*) in a 96-well plate. The fluorescence emission intensity (excitation: 340 nm, emission: 450 nm) of the sample was detected by a microplate reader (Thermo Scientific, Waltham, MA, USA) after the plate was incubated for 10 min at room temperature.

A series of different concentrations of tryptophan (0.01, 0.05, 0.10, 0.20, 0.50, and 0.60 mmol/mL) were used to establish the standard curve. The DH is calculated by the ratio of free amino groups after digestion to the total amino acid content of the protein. All determinations were performed in triplicate. 

According to Gulati et al. [21], the DH, or extent of proteolytic hydrolysis, was calculated using the following equation:DH (%) = (hs/htotal) × 100% (1)
where hs is defined as the mmol of free amino groups per gram of protein in the sample, and htotal is the mmol of free amino groups per gram of protein, assuming complete hydrolysis of the protein (6.99 mmol/g protein). All tubes (representing triple samples from above) were measured three times.

### 2.5. MALDI-TOF-MS Analysis of Sample Digest

Since the sample detected by MALDI-TOF-MS (Bruker, Bremen, Germany) needs to be desalted, the Zip tip C18 pipette tip was used to remove the salts of the samples. Ten microliters of digested products were absorbed by a C18 column pre-equilibrated with acetonitrile and 0.1% TFA, and salt was washed with 0.1% TFA, then the target peptides were eluted using a solvent of acetonitrile-water (20:80, *v*/*v*). 

One microliter of the desalted sample was spotted on the ground steel BC target (MTP 384, Bruker, Bremen, Germany). After drying at room temperature, 1 μL of the CHCA (α-Cyano-4-hydroxy cinnamic acid) matrix solution was added to the sample to be crystallized. MALDI-TOF mass spectra were performed on an UltrafleXtreme TOF-TOF mass spectrometer (Bruker, Bremen, Germany), operating in reflector positive ion mode. The detection m/z mass range was 0–5000, and the laser intensity was 60%. The external calibration was performed by Peptide Calibration Standard II (Bruker) every half an hour. The data were analyzed by the Flex Analysis Batch Process. 

### 2.6. Intrinsic Fluorescence Measurement of Sample

Fluorescence data were collected using LUMINA Fluorescence Spectrometer, (Thermo Scientific, Waltham, MA, USA). Intrinsic fluorescence of proteins is mainly attributed to tryptophan and tyrosine, which are particularly sensitive to the polarity of the microenvironment. For measuring intrinsic fluorescence, the samples prepared earlier were centrifuged at 8000× *g* for 5 min, the supernatants were excited at 280 nm, and the fluorescence spectrum was recorded from 295 to 400 nm in an interval of 5 nm.

### 2.7. Surface Plasmon Resonance (SPR) 

The surface plasmon resonance method was modified according to Xu et al. [22]. The HC200M sensor chip was thoroughly cleaned with 10 mM sodium acetate buffers at pH 4.5. The target proteins (myoglobin, myosin, and trypsin) were covalently immobilized on the flow cell surface of the sensor chip by the amine coupling method. The 1:1 (*v*/*v*) mixture of 0.4 M 1-ethyl-3-(3-dimethylaminopropyl) carbodiimide hydrochloride (EDC) and 0.1 M N-hydroxysuccinimide (NHS) was employed to activate the carboxyl groups on the flow cell surface for 7 min at a flow rate of 10 μL/min. Different proteins (50 μg/mL) were pulse injected onto the activated cell surface. About 1.0 M ethanolamine hydrochloride-NaOH (pH 8.5) was injected slowly for 7 min to block the remaining free activated ester groups on the surface of the sensor chip. 

The BIAcore T200 (Cytiva, Marlborough, MA, USA) sensorgrams (resonance units (RUs) versus time) were recorded when different concentrations of EGCG (0.1 to 1 mg/mL) were injected at a flow rate of 40 μL/min at 37 °C, to measure the binding kinetics. The flow times allowed for the association and dissociation of proteins and EGCG were 7 and 10 min, respectively. After each detection, the sensor chip was regenerated by 20 μL of 10 mM Glycine-HCl (pH 2.0).

### 2.8. Reactive Sulfhydryl Content Determination

The method of reactive sulfhydryl content was modified according to Chen et al. [23]. The samples were prepared as described in Section 2.2, then lyophilized and ground to powder. The weight of CP-TP for determination was calculated according to the 3 mg chicken protein to ensure the consistency of the reaction substrate. The sample was dispersed in 1 mL of 250 mM Tris-glycine buffer (pH 8.0) with 8M urea and 20 μL of Ellman’s reagent (4 mg/ mL DTNB configurated by 250 mM Tris-glycine buffer). The mixture was incubated in the dark at room temperature for 1 h and centrifuged at 8000× *g* for 10 min. The absorbance of the supernatant was measured at 412 nm, and the supernatant without sample was used as a control. Absorbance values were converted to amounts according to the following equation:C_SH_ = A/εb(2)
where A is the absorbance at 412 nm, ε is the extinction coefficient of 13,600 M^−1^ cm^−1^, and b is the cell path. The results are expressed in μmol per g of protein. The analysis was performed in triplicate.

### 2.9. Statistical Analysis

Statistical analysis of experimental data was carried out using SPSS22.0 software(International Business Machines Corporation, Almonk, NY, USA). Significant differences among experimental mean values including the digestibility, the binding constants of EGCG with different proteins, and reactive sulfhydryl group content were analyzed by one-way analysis of variance (ANOVA) coupled with Duncan’s test at a statistical significance level of 95% (*p* < 0.05).

## 3. Results and Discussion

### 3.1. The Changes of Chicken Protein Digestibility under Different Concentrations of Tea Polyphenol 

Different tea polyphenol concentrations (0–20 mg/mL) were selected to explore the effects of tea polyphenol on chicken protein in different modes of interaction. The results (Figure 1a) showed that the digestibility of CP-TP decreased with the increase in tea polyphenol concentration in three interaction modes, suggesting that the digestibility of chicken protein was inhibited by tea polyphenol. Moreover, comparing (n)CP-TP at different tea polyphenol concentrations to (h)CP-TP, the digestibility of (h)CP-TP was lower, indicating that the digestibility of chicken protein was also inhibited by heating. In addition, the digestibility of CP-TP(h) was lower than that of (h)CP-TP, which indicated that the inhibition effect of interaction between heated chicken protein and tea polyphenol at room temperature on chicken protein digestibility was weaker than that of co-heated chicken protein and tea polyphenol.

Different temperatures (20, 40, 60, 80, and 100 °C) were selected to research the effects of temperature on the chicken protein digestibility at four different tea polyphenol concentrations (0 mg/mL, 0.1 mg/mL, 1 mg/mL, and 10 mg/mL). As shown in Figure 1b, the digestibility of chicken protein decreased with the increase in heating temperature at different tea polyphenol concentrations, which illustrated that heating inhibits the digestibility of chicken protein, and the effects have a positive correlation with temperature. Furthermore, the CP-TP digestibility decreased with the increased concentration of tea polyphenol at different temperatures, indicating that tea polyphenol inhibited the digestibility of chicken protein at any temperature.

Protease, as a kind of protein, potentially interacts with polyphenols resulting in the enzyme activity decreasing. To investigate whether tea polyphenols affect the digestibility of chicken protein by interacting with proteases, pepsin and trypsin were selected to interact with tea polyphenols at different tea polyphenol addition levels (0 mg, 0.1 mg, 1 mg, and 10 mg). (a) The chicken protein was digested according to the simulated digestion process described in Section 2.3, except that the pepsin used in digestion was pre-mixed with tea polyphenol for 10 min at room temperature. (b) The chicken protein was digested according to the simulated digestion process described in Section 2.3, except that the trypsin used in digestion was pre-mixed with tea polyphenol for 10 min at room temperature. According to the results shown in Figure 1c, it was found that the chicken protein digestibility was decreased with the increase in tea polyphenol, which indicates that the interaction between tea polyphenol and proteases inhibited the chicken protein digestibility, and the inhibition positively related with tea polyphenol content. 

### 3.2. The Changes of Chicken Protein Digested Peptide Mapping at Different Tea Polyphenol Concentrations at Different Temperatures

The digested peptide mapping of CP-TP at different tea polyphenol concentrations was analyzed by MALDI-TOF, and the MS spectrum was shown in Figure 2. According to the peptide mapping, the peptides’ molecular weight of digested CP-TP and chicken protein is mainly concentrated in the range of 1000–2400 Da. The spectrum shows that the intensity of peptide from digested (n)CP-TP was higher than from digested (h)CP-TP and CP-TP(h). With the concentration of tea polyphenol increased, the peptide intensity and variety decreased, particularly, the peptide from the digested (n)CP-TP and (h)CP-TP concentrated to the lower molecular weight, while the peptide from digested CP-TP(h) concentrated to higher molecular weight. The results indicated that the peptide from digested CP-TP was related to the tea polyphenol concentration and the interaction modes. Lower-concentration tea polyphenol induced more peptides, and chicken protein heated with tea polyphenol has more influence on protein digestion. 

Coincidentally, the peptide molecular weight from digested CP-TP and chicken protein at different temperatures was also mainly concentrated in the range of 1000–2400 Da (Figure 3), which suggested that temperature variation did not change the peptide molecular weight. The peptide variety and intensity from digested CP-TP were less than the chicken protein at the same temperature, and decreased with the increase in tea polyphenol concentration, indicating that tea polyphenol affected the chicken protein digestion to produce peptide at any temperature, and the variation trends in peptide variety and intensity were the same as the digestibility. Especially, when the tea polyphenol concentration was 10 mg/mL, the peptide mapping at different temperatures did not change significantly, which suggested that high-concentration tea polyphenol had a great inhibition effect on the chicken protein digestion, even without a heating assistant.

### 3.3. The Changes of Chicken Protein Intrinsic Fluorescence

According to the previous experimental results, tea polyphenols had a great inhibition effect on chicken protein digestion. It was essential to research the structure of CP-TP to investigate the inhibition mechanism. Tryptophan and tyrosine residues are the main intrinsic fluorescent groups of proteins, and the variations in protein intrinsic fluorescence intensity and emission peak are closely related to their microenvironments [24], so the intrinsic fluorescence spectrum was employed to explore the protein conformation changes.

The intrinsic fluorescence of CP-TPs at different interaction modes was detected, and the results were shown in Figure 4. A significant blue shift in the emission wavelength was observed in the CP-TPs at different interaction modes, indicating that the protein conformation was more compact after adding tea polyphenol, since the tryptophan inside the protein molecule had a shorter fluorescence emission wavelength than on the surface of the protein molecule [25]. Furthermore, because tryptophan was one of the cleavage sites of digestive enzymes, compact protein conformation reduced the contact between cleavage sites and proteases [26], which explained the reduced digestibility of chicken protein when the tea polyphenol was added. However, when the concentration of the tea polyphenol increased, the emission wavelength of CP-TPs red-shifted. According to previous studies, polyphenols dissolved in solvents could increase the polarity of the solution [27], while the fluorescence emission peak of tryptophan would redshift with the increase in the polarity of the solvent [28]. Therefore, the redshift of emission wavelength caused by the increase in polyphenol concentration was ascribed to the increased polarity of the solution. When the concentration of the tea polyphenol interacted with chicken protein was 0–2 mg/mL, the fluorescence intensity of CP-TPs increased with the increase in the tea polyphenol concentration. When tryptophan is located inside the protein in a hydrophobic environment, the fluorescence quantum yield is high, and the fluorescence intensity increases [12]. Therefore, the increasing addition of tea polyphenol to chicken protein led to an increase in fluorescence intensities, indicating that the structure of the chicken protein became more compact after interacting with the tea polyphenol, which also explains the decreasing digestibility of CP-TP with the increasing tea polyphenol concentration. When the concentration of tea polyphenol was 5–20 mg/mL, the fluorescence intensity of CP-TP decreased with the increasing tea polyphenol concentration. There were two possible reasons: one was protein unfolding, and tryptophan was exposed to a hydrophilic environment, which led to the increase in collision energy transfer around tryptophan and reduced the fluorescence quantum yield and fluorescence intensity [29]; another reason was that the protein structure became more compact, and tryptophan was encapsulated inside the protein and cannot be excited to generate fluorescence. According to Burstein et al. [25], when the fluorescence intensity decreases, the fluorescence emission wavelength undergoes a red shift, indicating that tryptophan was on the surface of the protein. Therefore, when the tea polyphenol concentration increased, the decrease in fluorescence intensity was due to the loose protein structure and the exposure of tryptophan to the hydrophilic environment. However, when tryptophan was exposed to solution, there should be more cleavage sites to react with digestive enzymes and the digestibility of CP-TP should increase, which was contrary to experimental results. 

Since proteases also affected protein digestion, the intrinsic fluorescence of the complex of tea polyphenol and protease was determined (Figure 5). It was found that the fluorescence intensity of both digestive enzymes decreased with the tea polyphenol concentration increasing, indicating that the tea polyphenol could interact with pepsin and trypsin, causing changes in protease structure and activity. In this experiment, tea polyphenol was interacted with chicken protein before digestion, so the tea polyphenol preferred to combine with chicken protein. At high concentrations of tea polyphenol, there may be tea polyphenol that did not bind to chicken protein, and this tea polyphenol could bind to proteases during digestion. This combination can cause a decrease in protease activity, inducing a decrement in chicken protein digestibility. Moreover, at a high concentration of tea polyphenol, the loose tertiary structure of chicken protein may expose more digestion sites and increase the digestibility of chicken protein. However, our experimental results showed that the chicken protein digestibility was lower at a high concentration of tea polyphenol. Therefore, when tea polyphenol concentration was 5–20 mg/mL, we believed that the main reason for the decrease in digestibility was the activity decrease in protease caused by the combination of tea polyphenol and proteases. 

In addition, the intrinsic fluorescence of CP-TP at different temperatures was studied (Figure 6), and it was found that the emission wavelength of CP-TP did not change with the increase in temperature at the same tea polyphenol concentration, which showed that the temperature had no effect on the emission wavelength. With the increase in temperature, the fluorescence intensity decreased gradually, mainly because of protein aggregation caused by heating [30]. The protein aggregation induced some tryptophan to be hidden, which caused the decrement in fluorescence intensity. The variation trend of fluorescence intensity under different temperatures was consistent with the chicken protein digestibility, suggesting that the effect of temperature on the digestibility of protein was mainly caused by the hiding of digestibility sites induced by protein aggregation.

### 3.4. The Binding Constants of EGCG with Different Proteins

The binding kinetics between proteins (chicken protein and protease) and tea polyphenols were measured to further analyze the interaction of tea polyphenols with chicken protein and protease. EGCG is the main component of tea polyphenols, myosin and myoglobin are the main components of chicken protein, and trypsin is the main protease. Therefore, the binding parameters of the three proteins (myoglobin, myosin, and trypsin) with EGCG are given in Table 1, including the association rate constant (Ka, M^−1^ s^−1^) and dissociation (Kd, s^−1^) rate constant and equilibrium dissociation constant (KD, M^−1^). The Ka values indicated that the binding of EGCG to myoglobin and myosin was more rapid than trypsin. Similarly, the KD values indicated that EGCG dissociated from trypsin more quickly than myoglobin and myosin. According to the result, tea polyphenols are preferentially bound with chicken protein, affecting the structure and the digestibility of the chicken protein in the digestive system. With the increase in tea polyphenol concentration, the binding between tea polyphenols and chicken protein saturates, and the remaining tea polyphenols could combine with trypsin, affecting both the structure of chicken protein and proteases and also the digestion of chicken protein. The results were consistent with fluorescence results.

### 3.5. The Changes of Chicken Protein Reactive Sulfhydryl Contents

The change of reactive sulfhydryl groups in the protein is a persuasive indicator to evaluate the conformation variation of the protein. The reactive sulfhydryl groups of chicken protein and CP-TP were detected. The content of the reactive sulfhydryl group in CP-TP gradually decreased with the increase in polyphenol concentration (0–2 mg/mL) in different interaction modes (Table 2). According to the reported research [31,32], polyphenols were easily oxidized into quinones and reacted with reactive sulfhydryl groups of proteins to form covalent complexes. Therefore, when the concentration of the tea polyphenol increases (0–2 mg/mL), the covalent binding of sulfhydryl groups with polyphenols resulted in a decrease in reactive sulfhydryl groups. When the concentration of the tea polyphenol increased from 5 mg/mL to 20 mg/mL, the reactive sulfhydryl content of chicken protein increased. It has been reported that green tea extract as a natural phenolic antioxidant could disrupt the disulfide network in meat protein [33]. Therefore, the covalent bonding saturates between tea polyphenol and protein under higher concentrations of tea polyphenol, and the excess tea polyphenol could reduce the disulfide bond, resulting in the increase in the reactive sulfhydryl content. Compared with the (n)CP-TP, the reactive sulfhydryl content of (h)CP-TP was significantly lower, suggesting that heating induced the reactive sulfhydryl to decrease. Heating contributed to facilitating the formation of disulfide bonds [34,35]. Therefore, the decrease in reactive sulfhydryl content was caused by the formation of disulfide bonds induced by heating. With the increase in tea polyphenol content, the difference gradually decreased between the (n)CP-TP and the (h)CP-TP in reactive sulfhydryl content. The results indicated that as the tea polyphenol concentration increased, the covalent binding between the tea polyphenol and native chicken protein also increased, which offset the decrease of reactive sulfhydryl content in heated chicken protein caused by disulfide bond formation induced by heating. When the tea polyphenol concentration was 5 mg/mL, there was no significant difference in reactive sulfhydryl content between the (n)CP-TP and the (h)CP-TP, suggesting that the reactive sulfhydryl content reached a balance. In this condition, the balance was obtained between the disulfide bond increase caused by heating and the reactive sulfhydryl group decrease caused by the tea polyphenol covalently binding. Consequently, there was no significant difference in the reactive sulfhydryl content.

The variation of reactive sulfhydryl groups in the CP-TP at different temperatures was analyzed, and the results are given in Table 3. When the chicken protein was heated at different temperatures, the reactive sulfhydryl content decreased with the temperature increasing. However, comparing the reactive sulfhydryl content of CP-TP at different concentrations of tea polyphenol at the same temperature, it was found that the reactive sulfhydryl content of CP-TP among the different concentrations of tea polyphenol varied significantly. This indicated that the effect of disulfide bond formation induced by heating on the decrement of reactive sulfhydryl content was weaker than that of covalent binding between tea polyphenol and chicken protein [31]. It suggested that the activity of the covalent binding reaction between the sulfhydryl group and polyphenol was stronger than that of the oxidation between the sulfhydryl groups to form a disulfide bond [35]. At each temperature, when the concentration of tea polyphenol was 10 mg/mL, the reactive sulfhydryl content of CP-TP was higher than that of 1 mg/mL. The results indicated that the interaction between tea polyphenol and chicken protein at different temperatures would prefer covalently bind to reactive sulfhydryl than reduce disulfide bonds when the concentration of tea polyphenol increased.

## 4. Conclusions

In this paper, the effects of tea polyphenols on chicken protein digestion were studied, and it was found that tea polyphenols and heating inhibited chicken protein digestion. In addition, the chicken protein digestibility had a negative correlation with the tea polyphenol concentration and interaction temperature. Also, the digestibility of the complex of tea polyphenol and chicken protein at different modes was in the order of (n)CP-TP>(h)CP-TP>CP-TP(h). The changes in chicken protein digested peptide mapping showed that the inhibition variation trends in peptide variety and intensity were the same as the digestibility. 

At low concentrations of tea polyphenols, the conformation of CP-TP became compact, which was induced by covalent interaction between the reactive sulfhydryl group and tea polyphenol and non-covalent interaction including electrostatic interaction and hydrogen bond change. The structure changes hid more protease cleavage sites and resulted in lower digestion performance. At high concentrations of tea polyphenols (5–20 mg/mL), the disulfide bonds in chicken protein were reduced by tea polyphenols, inducing a loose protein conformation. But the excessive tea polyphenol would interact with digestive enzymes, and the digestion inhibition was mainly caused by decreased digestive enzyme activity. 

In this study, the effects of tea polyphenols on chicken protein digestion and the mechanism under thermal processing were researched. However, there are still some limitations, including the protein digestion changes under different interaction systems and the changes in the properties of tea polyphenols during the interaction. For better investigation of the interaction between protein and tea polyphenol, further research is required in the future.

## Figures and Tables

**Figure 1 foods-12-02905-f001:**
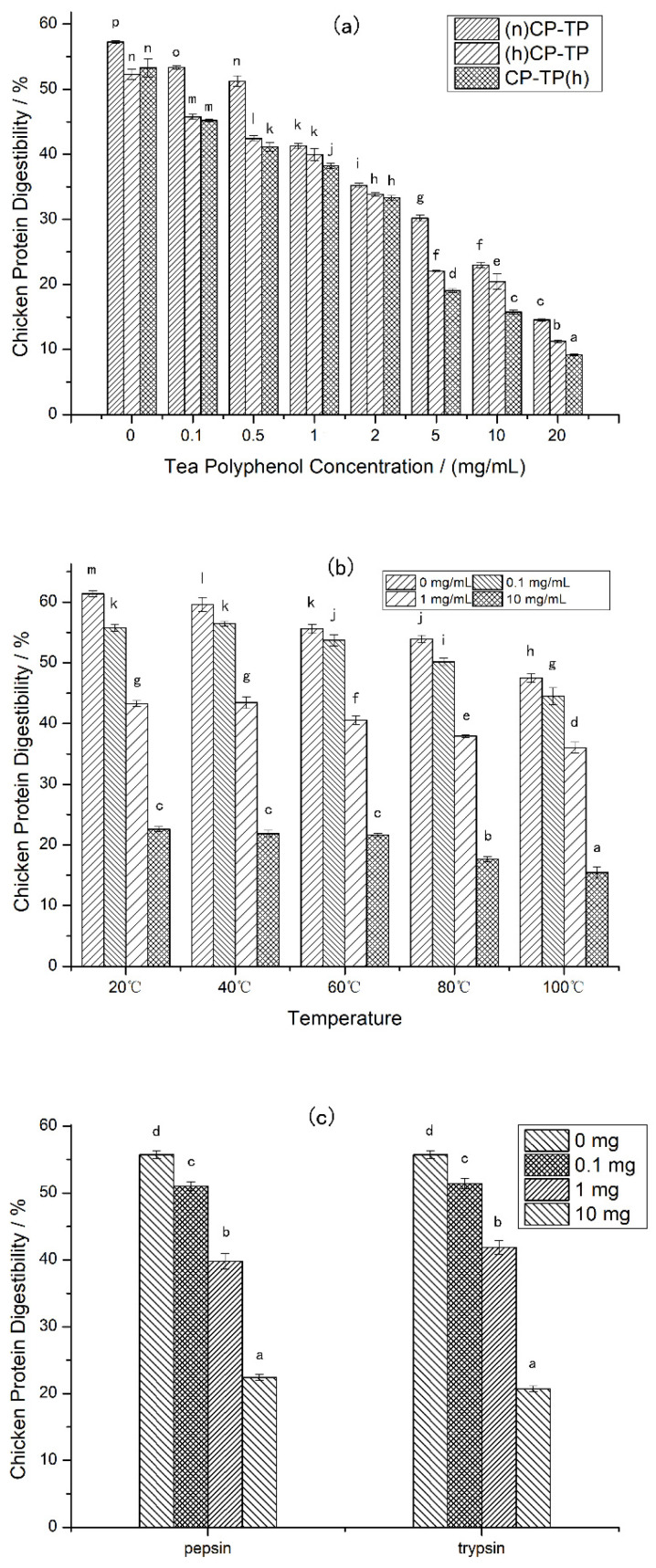
The digestibility of CP-TP and chicken protein at different conditions. (**a**) The digestibility of CP-TP at different interaction modes ((n)CP-TP (complex of native chicken protein and tea polyphenol), (h)CP-TP (complex of heated chicken protein and tea polyphenol), and CP-TP(h) (complex of chicken protein and tea polyphenol heated together)). (**b**) The digestibility of chicken protein and CP-TP at different temperatures. (**c**) The digestibility of chicken protein under the interaction between tea polyphenol and proteases. The different lowercase letters represent significant differences between the data, and the same lowercase letters represent no significant differences between the data in the same figure (*p* < 0.05).

**Figure 2 foods-12-02905-f002:**
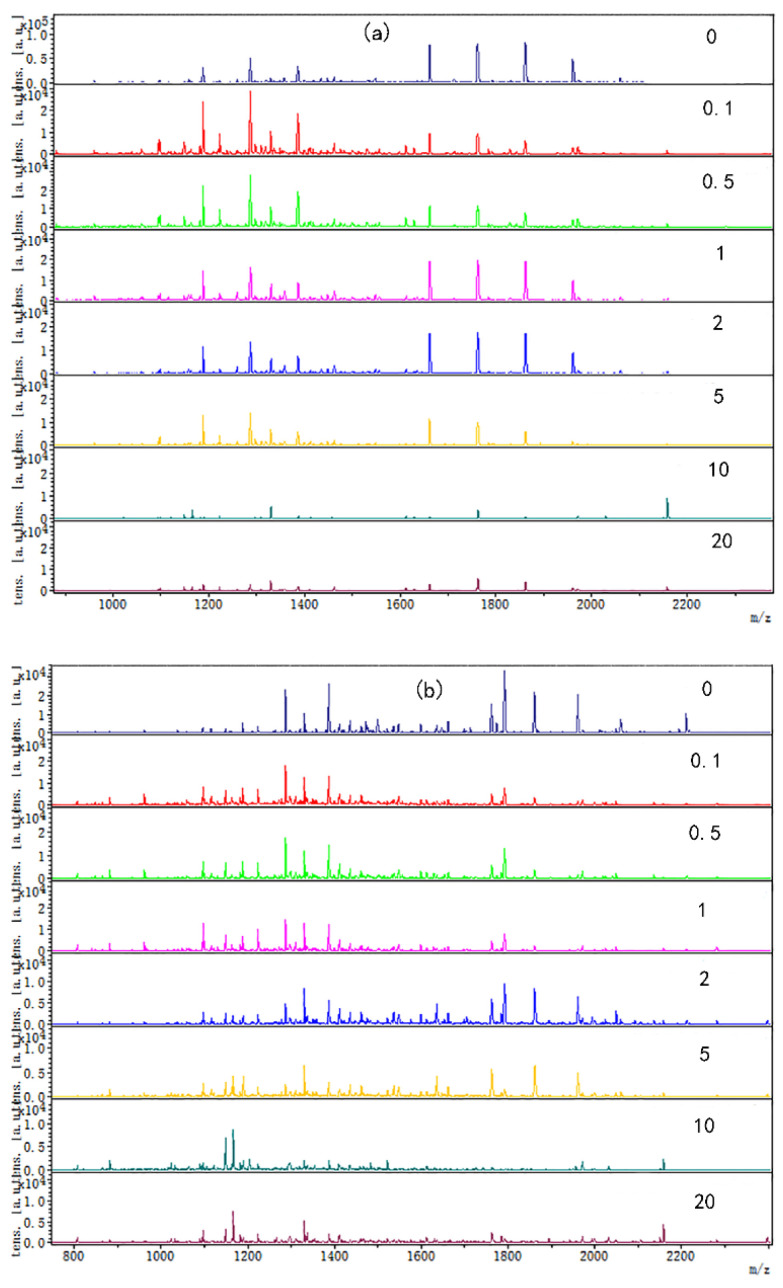
The digested peptides MS spectrum of CP-TP treated at different interaction modes: (**a**) (n)CP-TP (complex of native chicken protein and tea polyphenol), (**b**) (h)CP-TP (complex of heated chicken protein and tea polyphenol), and (**c**) CP-TP(h) (complex of chicken protein and tea polyphenol heated together).

**Figure 3 foods-12-02905-f003:**
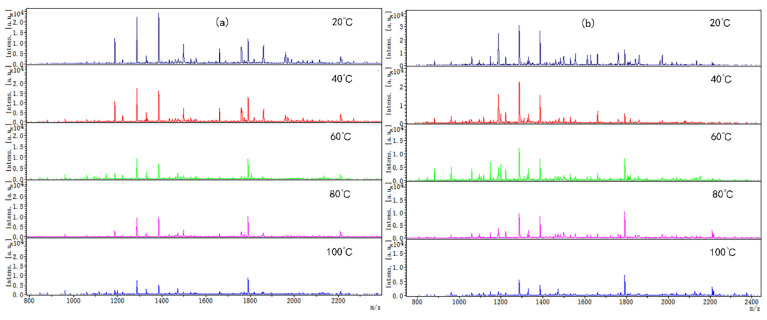
The digested peptides MS spectrum of chicken protein and CP-TP treated at different temperatures. (**a**) Digested peptides of chicken protein treated at different temperatures. (**b**) Digested peptides of CP-TP at 0.1 mg/mL of tea polyphenol treated at different temperatures. (**c**) Digested peptides of CP-TP at 1 mg/mL of tea polyphenol treated at different temperatures. (**d**) Digested peptides of CP-TP at 10 mg/mL of tea polyphenol treated at different temperatures.

**Figure 4 foods-12-02905-f004:**
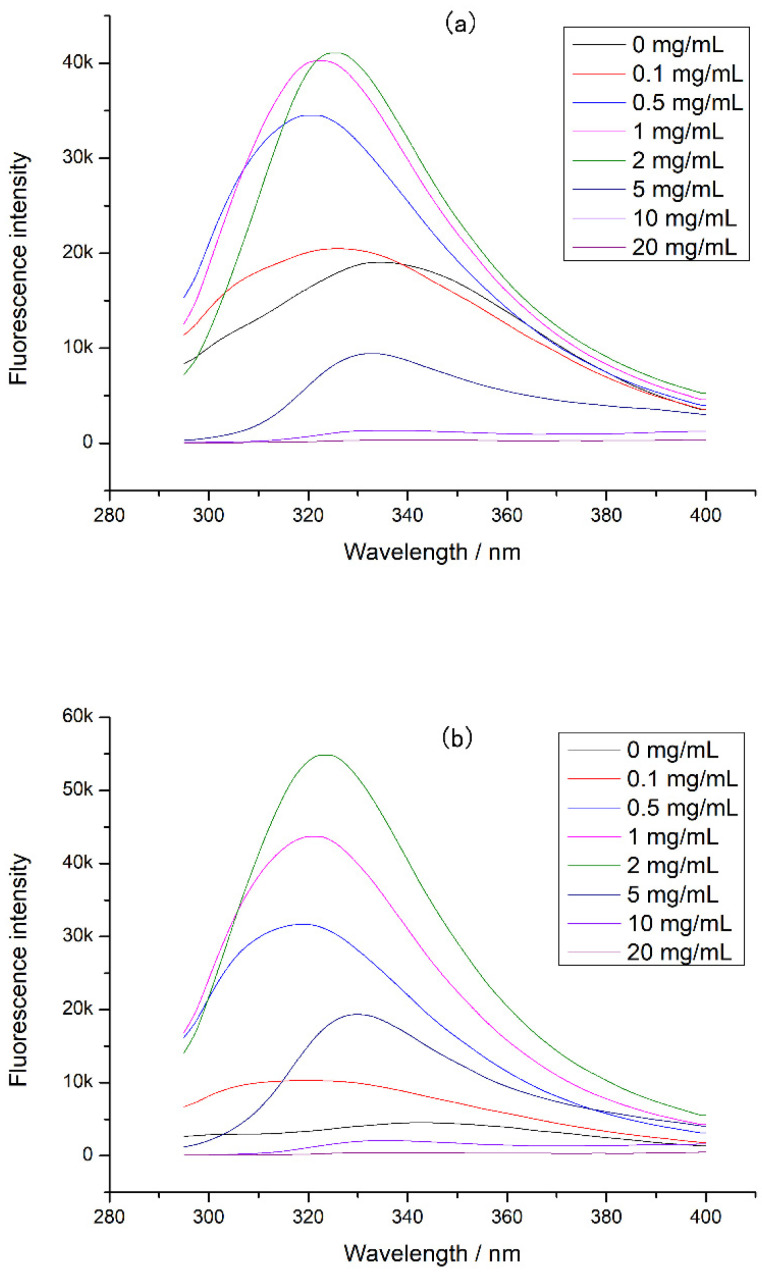
The intrinsic fluorescence of CP-TP at different interaction modes: (**a**) (n)CP-TP (complex of native chicken protein and tea polyphenol), (**b**) (h)CP-TP (complex of heated chicken protein and tea polyphenol), and (**c**) CP-TP(h) (complex of chicken protein and tea polyphenol heated together).

**Figure 5 foods-12-02905-f005:**
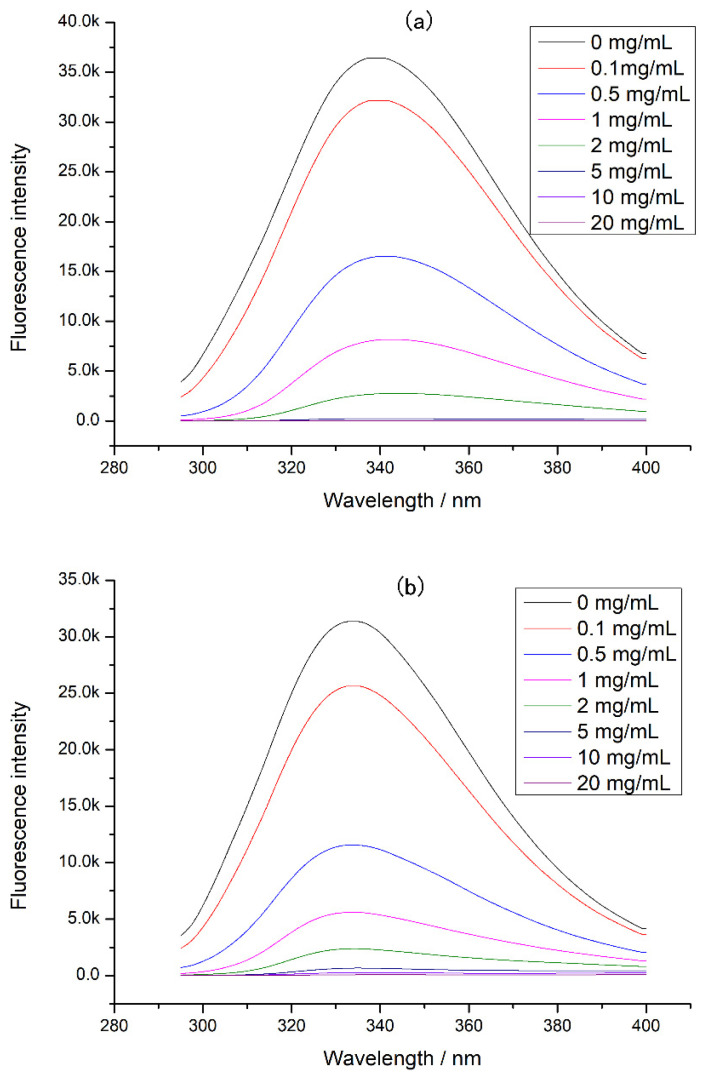
The intrinsic fluorescence of protease at different tea polyphenol concentrations: (**a**) pepsin at different tea polyphenol concentrations and (**b**) trypsin at different tea polyphenol concentrations.

**Figure 6 foods-12-02905-f006:**
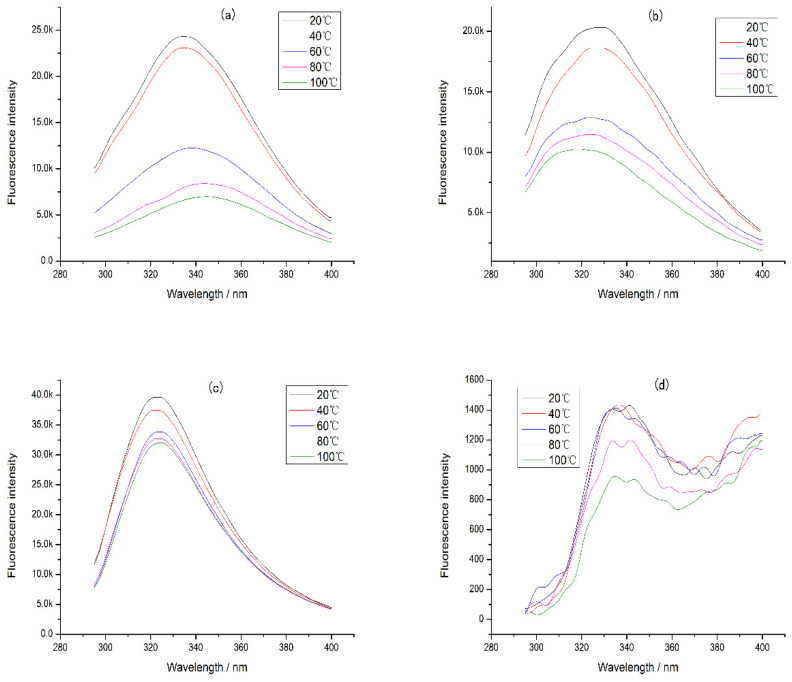
The intrinsic fluorescence of chicken protein and CP-TP at different temperatures. (**a**) Chicken protein treated at different temperatures. (**b**) CP-TP at 0.1 mg/mL of tea polyphenol treated at different temperatures. (**c**) CP-TP at 1 mg/mL of tea polyphenol treated at different temperatures. (**d**) CP-TP at 10 mg/mL of tea polyphenol treated at different temperatures.

**Table 1 foods-12-02905-t001:** The binding constants of EGCG with different proteins.

Sample	Ka (M^−1^ s^−1^)	Kd × 10^−4^ (s^−1^)	KD × 10^−7^ (M)
Myoglobin	1514.0 ± 79.8 ^b^	1.93 ± 0.07 ^a^	1.28 ± 0.14 ^a^
Myosin	3051.0 ± 46.0 ^c^	4.79 ± 0.43 ^c^	1.57 ± 0.02 ^a^
Trypsin	376.6 ± 6.2 ^a^	2.73 ± 0.20 ^b^	7.24 ± 0.24 ^b^

Values are displayed as average ± standard deviation values followed by different lowercase letters. The different lowercase letters are significantly different in the same column, calculated according to ANOVA and Duncan’s test (*p* < 0.05).

**Table 2 foods-12-02905-t002:** The reactive sulfhydryl contents of CP-TP at different interaction modes ((n)CP-TP (complex of native chicken protein and tea polyphenol), (h)CP-TP (complex of heated chicken protein and tea polyphenol), CP-TP(h) (complex of chicken protein and tea polyphenol heated together)).

TP Concentration	(n)CP-TP	(h)CP-TP	CP-TP(h)
0 mg/mL	93.76 ± 3.83 ^l^	78.6 ± 2.55 ^j^	78.98 ± 3.49 ^j^
0.1 mg/mL	87.8 ± 1.54 ^k^	62.15 ± 2.22 ^h^	73.22 ± 3.22 ^i^
0.5 mg/mL	34.71 ± 0.08 ^fg^	24.66 ± 0.24 ^d^	28.51 ± 0.32 ^e^
1 mg/mL	20.34 ± 0.45 ^c^	17.31 ± 0.15 ^a^	17.74 ± 0.33 ^ab^
2 mg/mL	17.34 ± 0.17 ^a^	15.59 ± 0.21 ^a^	16.29 ± 0.47 ^a^
5 mg/mL	20.74 ± 0.40 ^c^	19.87 ± 0.60 ^bc^	20.2 ± 0.56 ^c^
10 mg/mL	26.96 ± 0.92 ^de^	27.19 ± 0.59 ^e^	24.56 ± 0.48 ^d^
20 mg/mL	37.06 ± 0.61 ^g^	34.75 ± 0.65 ^fg^	33.38 ± 1.53 ^f^

Values are displayed as average ± standard deviation values (μmol/g protein) followed by different lowercase letters. The different lowercase letters are significantly different in the same table, calculated according to ANOVA and Duncan’s test (*p* < 0.05).

**Table 3 foods-12-02905-t003:** The reactive sulfhydryl contents of chicken protein and CP-TP at different temperatures.

TP Concentration	20 °C	40 °C	60 °C	80 °C	100 °C
0 mg/mL	100.38 ± 0.92 ^k^	91.55 ± 1.92 ^j^	90.24 ± 1.43 ^ij^	85.53 ± 2.99 ^g^	79.81 ± 2.05 ^f^
0.1 mg/mL	89.21 ± 1.36 ^hi^	90.15 ± 1.91 ^ij^	86.59 ± 2.55 ^g^	87.5 ± 2.19 ^gh^	72.85 ± 2.37 ^e^
1 mg/mL	20.34 ± 0.29 ^b^	20.44 ± 0.43 ^b^	19.52 ± 0.34 ^ab^	18.72 ± 0.13 ^ab^	17.74 ± 0.25 ^a^
10 mg/mL	26.96 ± 0.60 ^d^	26.06 ± 1.55 ^cd^	25.91 ± 0.83 ^cd^	24.99 ± 0.40 ^cd^	24.56 ± 0.37 ^c^

Values are displayed as average ± standard deviation values (μmol/g protein) followed by different lowercase letters. The different lowercase letters are significantly different in the same table, calculated according to ANOVA and Duncan’s test (*p* < 0.05).

## Data Availability

The data used to support the findings of this study can be made available by the corresponding author upon request.

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
