# Peer review of "The Effects of Tea Polyphenol on Chicken Protein Digestion and the Mechanism under Thermal Processing"

_foods, 2023, doi:10.3390/foods12152905_

Round 1

Reviewer 1 Report

The article uses a variety of techniques to explore the effect of tea polyphenols on the digestion of chicken protein, and the structure is clear. Below are some of the technical and non-technical points which should be consider:

1. Part 3.2 only provides an intuitive description of the figure, lacks relevant explanations. More mechanistic explanations should be added to give reviewers a better understanding.

2. The figures are not very clear, and higher-resolution pictures should be provided, especially figure 1, 4, 6.

3. L293 TP-CPs?

4. L381:I think "The different lowercase letters are significantly difference in same column/line"

5. The MS mainly discusses the effect of adding tea polyphenols on the digestion of chicken protein, but does not mention tea polyphenols behavious in the digestion process. Tea polyphenols are also known to be beneficial to health. Does this study understate the potential benefits of tea polyphenols? And what might happen to tea polyphenols during digestion? 

6. The conclusion section summarizes the experimental results. However, there is a lack of thinking about limitations, I think this research also has limitations, maybe some critical content can be supplemented, including limitations and considerations for future research.

please check the grammar and the logic of the text, for example the introduction should be more concentrated.

Author Response

We appreciate for your warm work earnestly, and hope that the correction will meet with approval.

  1. Part 3.2 only provides an intuitive description of the figure, lacks relevant explanations. More mechanistic explanations should be added to give reviewers a better understanding.

Response:

Thank you for your comments. We have investigated a lot of literatures when writing this part. Unfortunately, there isn’t reference about the peptide profile of digested chicken protein detected by MALDI-TOF. Therefore, the relevant results are briefly introduced. This part was researched to investigate the effects of tea polyphenols on the peptide molecular weight of digested chicken protein and support the results of part 3.1. We believe that the change of digestion was caused by protein structure change and interaction with proteases. The mechanism explanations for the effects were discussed in the following part of the protein structural analysis. For example, at L289-292, “Furthermore, because tryptophan was one of the cleavage sites of digestive enzymes, compact protein conformation reduced the contact between cleavage sites and proteases [25], which explained the reduced digestibility of chicken protein when the tea polyphenol was added.”.

  1. The figures are not very clear, and higher-resolution pictures should be provided, especially figure 1, 4, 6.

Response:

Thank you for your suggestion. We are sorry for the unclear pictures, and we have uploaded new pictures with higher resolution.

  1. L293 TP-CPs?

Response:

We're sorry for the writing, which was a careless mistake. It should be CP-TPs. We have corrected this mistake in manuscript which is marked red.

  1. L381:I think "The different lowercase letters are significantly difference in same column/line".

Response:

Thank you for your comments. It should be “same column/line”. We have made a correction in the manuscript which is marked in red. Thank you again for pointing our problem.

  1. The MS mainly discusses the effect of adding tea polyphenols on the digestion of chicken protein, but does not mention tea polyphenols behavious in the digestion process. Tea polyphenols are also known to be beneficial to health. Does this study understate the potential benefits of tea polyphenols? And what might happen to tea polyphenols during digestion?

Response:

Thank you for your comments. The main subject of this paper is to study the effect of tea polyphenols on the digestion and structure of chicken protein during thermal processing. The MALDI-TOF was selected to analyze the peptide profile of chicken protein. However, MALDI-TOF is suitable for the analysis of biological macromolecules such as peptides, lipids, carbohydrates and other organic macromolecules. The micro-molecules such as tea polyphenols are subject to a lot of interference in the MALDI-TOF MS spectrum. Therefore, the MS results of this experiment cannot be used for the analysis of tea polyphenols. Tea polyphenols are known to be beneficial to health, but it’s not the mainly object of this paper. The effect of protein-tea polyphenols interaction on tea polyphenols is also an interesting subject. We will investigate it in our future research.

  1. The conclusion section summarizes the experimental results. However, there is a lack of thinking about limitations, I think this research also has limitations, maybe some critical content can be supplemented, including limitations and considerations for future research.

Response:

Thank you for your valuable comments. Indeed, there are still some limitations in our current study, including the protein digestion changes under different interaction systems and the changes in the properties of tea polyphenols during the interaction. These are the contents that we will research in the future, and we will continuously complete our research system. The limitations and considerations for future research was added into the conclusion section, which marked in red.

Once again, thank you very much for your comments and suggestions.

Reviewer 2 Report

The main topic is the influence of green tea polyphenols on the digestion of chicken meat proteins under different temperature conditions and at different concentrations of polyphenols.

The topic is relevant to the field. In WoS there is an article "Comparative study on the in vitro digestibility of chicken protein after different modifications. Chen, J.H. et al. DOI 10.1016/j.foodchem.2022.132652”, which complements this study. The other articles do not have a common theme with this study.

The article extends the research topic to other areas that have not been published.

The article is logical from the point of view of methodology and there is no need to supplement or modify it.

References are appropriate for this topic. Reference No. 18 (lines 510-518) is a self-citation.

In the description of the tables and graphs, it would be advisable to add an explanation of the abbreviations used. These abbreviations are explained in the text, but using a separate table or graph will not make it clear what they mean.

Thank you.

Author Response

Thank you for your suggestion. In the newly revised version, the explanation of the abbreviations has been added to the relevant table and figure notes. Please review it to see if it is appropriate.

Reviewer 3 Report

The manuscript can be reconsidered after major revision.

English quality should be improved.

Author Response

Thank you for your comments concerning our manuscript entitled “The effects of tea polyphenol on the chicken protein digestion and the mechanism under thermal processing”. For your convenience, the comment has been written in bold and revised portion are marked in red in the paper. The main corrections in the paper and the responds to the comments are as flowing:

  1. Abstract:

Line 23-25: The sentence should be revised.

The application of research has not been mentioned.

Response:

Thank you for your suggestion. We have rewritten this part which was marked in red. Please review it to see if it is appropriate.

  1. Introduction:

Line 65: to analyze the affinity………

Are there any previous studies on the interactions between chicken proteins and polyphenols? Please include them.

What would be the potential applications of these interactions?

Response:

We're sorry for the writing, which was a careless mistake. We have corrected this mistake in manuscript which is marked in red. According to your suggestion, we searched literatures, and found that there are previous studies on the interactions between chicken proteins and polyphenols. We have added them at Line 57 and marked in red.

The potential application of this study was that revealed the effect and the mechanism of polyphenol on the digestion performance of meat protein and provide more references for the further application of polyphenols in meat processing, which was added at abstract.

  1. Materials and methods

3.1 Line 82: chicken protein((n)CP)????????.

Response:

The full term is native chicken protein. There are different modes to process chicken protein in this paper. We named the unprocessed chicken protein as native chicken protein. In order to more concise expression in the paper and figures, we abbreviated it as (n)CP.

3.2 Line 83: to obtain the heated………

Response:

The full term is “to obtain the heated chicken protein”. We named the chicken protein heated at 100℃ for 10 min as heated chicken protein and abbreviated it as (h)CP.

3.3 Line 111: the DH (degree of hydrolysis)……..should be the degree of hydrolysis (DH)

Line 139: onto the sample to be crystallized……

Line 152: Surface…….

Response:

We're sorry for the writing, which was a careless mistake. We have corrected these mistakes in manuscript which are marked in red.

  1. Results and discussion

4.1 Section 3.1: Why do the digestibility of chicken proteins reduced with increasing polyphenol content. There should be reasons explained in this section.

Response:

Thank you for your comment. We believe that the change of digestion was caused by the change of protein structure and interaction with proteases. Therefore, the relevant results are briefly introduced. The explanations for the effects were discussed in the following part of the protein structural analysis. For example, at L289-292, “Furthermore, because tryptophan was one of the cleavage sites of digestive enzymes, compact protein conformation reduced the contact between cleavage sites and proteases [25], which explained the reduced digestibility of chicken protein when the tea polyphenol was added.”.

4.2 Line 245: higher molecular weight……..

Line 391: increasing should be increases.

Line 397: resulting in the reactive sulfhydryl content increasing? This should be revised as resulting in the increase of the reactive sulfhydryl content”.

Line 401: the decrease of reactive sulfhydryl content……..

Line 405: concentration increases

Response:

We're sorry for the writing, which was a careless mistake. We have corrected these mistakes in manuscript which are marked red.

4.3 Line 256-257: These sentences are not needed as they are part already mentioned in the method section.

We're sorry for the repetition. We have deleted these sentences.

4.4 Section 3.4: No references have been used to justify the results.

We have searched literatures, there are references to analyze the binding constants between enzymes or other proteins and small molecules through the surface plasmon resonance (SPR), such as “Revealing the mechanisms of starch amylolysis affected by tea catechins using surface plasmon resonance”, “Insights into protein-curcumin interactions: Kinetics and thermodynamics of curcumin and lactoferrin binding”, “Thermodynamic and kinetic study of epigallocatechin-3-gallate-bovine lactoferrin complex formation determined by surface plasmon resonance (SPR): A comparative study with fluorescence spectroscopy”. This experiment was designed with reference to these literatures, and SPR was applied to the affinity analysis between chicken protein, protease and tea polyphenols. However, there is no literature to analyze the affinity between these proteins mentioned in this study and tea polyphenols. According to the binding constants, the affinity of myosin and myoglobin to tea polyphenols was higher than that of trypsin. The results supported our hypothesis that tea polyphenols preferentially bound with chicken protein, and when the binding between tea polyphenols and chicken protein saturated, and the remaining tea polyphenols could combine with trypsin. Also, the results were consistent with fluorescence results. 

4.5 Line 412: Consequently, the reactive sulfhydryl content was no significant difference??? The sentence seems incorrect?

We're sorry for the writing. We have deleted the sentence, considering that it repeated with last sentence “When the tea polyphenol concentration was 5 mg/mL, there was no significant difference in reactive sulfhydryl content between the (n)CP-TP and the (h)CP-TP”.

4.6 Line 418-433: No literature studies have been cited to support the results?

Thank you for your valuable comments. According to your suggestion, we have added the relevant literatures to improve the discussion. Please review it to see if it is appropriate.

  1. Conclusions:

5.1 Line 450: At high-concentration of tea polyphenols…..

We're sorry for the writing, which was a careless mistake. We have corrected this mistake in manuscript which is marked in red. Similarly, we found the same mistake at first sentence of this paragraph, which also has been corrected and marked in red. Thank you for pointing out our problem.

5.2 Future work is not suggested in this section. There should be clearly shown the future work.

Thank you for your comments. There are still some limitations in our current study, including the protein digestion changes under different interaction systems and the changes in the properties of tea polyphenols during the interaction. These are the contents that we will continue to research in the future. The considerations for future research were added into the conclusion section, which marked in red. Please review it to see if it is appropriate.

 We appreciate for your warm work earnestly, and hope that the correction will meet with approval.

Round 2

Reviewer 1 Report

The paper has been revised accordingly and can be published. 

No more major revisions of language must be revised from my side. 

Reviewer 3 Report

The manuscript has been improved. It could be accepted.

English quality has been improved.